# Position: AI Should Facilitate Democratic Deliberation at Scale

**José Ramón Enríquez** [1 2]  **Jiaxin Pei** [2]  **Alex Pentland** [2 3]

## Abstract

AI systems can strengthen democracy by supporting deliberation at scale by addressing cognitive, social, platform-design, and market-driven frictions, while preserving human agency. Unlike proposals such as liquid democracy that restructure representation through vote delegation, in this position paper, we argue that AI-assisted deliberation offers a more promising path by lowering barriers to meaningful engagement without substituting machine judgment for human choice. Drawing on evidence from online deliberation platforms and experimental research, we identify four guiding principles: preserving agency and autonomy, encouraging mutual respect, promoting equality and inclusiveness, and augmenting rather than substituting active citizenship. We also address critical challenges, including alignment, sycophancy, training bias, and over-reliance on AI systems. We call on the machine learning community to develop deliberation-focused AI systems evaluated not on engagement metrics but on their capacity to facilitate informed, representative, and friction-robust discourse.

## 1. Introduction

Democratic societies face a representation crisis. Trust in government has declined across established and developing democracies, with citizens perceiving widening gaps between public preferences and policy outcomes (Citrin and Stoker, 2018; Valgarðsson et al., 2025; Wike, 2025). Affective polarization, the tendency to view political opponents with contempt rather than mere disagreement, has intensified (Reiljan et al., 2024; Boxell et al., 2024) and populist discourse has deepened (Guriev and Papaioannou, 2022). Social media platforms, initially celebrated as de-mocratizing forces that could amplify previously unheard voices (Farrell, 2012; Tudoroiu, 2014; Jost et al., 2018), have instead contributed to these problems: amplifying polarization, accelerating the spread of misinformation, and systematically rewarding divisiveness over civic engagement (Tucker et al., 2018; Bail et al., 2018; Vosoughi et al., 2018; Levy, 2021; Nyhan et al., 2023). However, at this transformative moment for generative foundation models, the question is not whether technological intervention is needed, but what form it should take.

We argue that AI systems, when designed to complement rather than substitute human judgment, offer an unprecedented opportunity to strengthen democracy by enabling deliberation at scale while addressing frictions that undermine meaningful civic engagement. This position draws on a rich tradition of democratic theory—from Habermas's communicative action (1984) through Landemore's open democracy (2020) to Mansbridge's systemic approach (2012)—which holds that legitimate collective decisions emerge through reasoned exchange among free and equal citizens. Our position contrasts with alternative proposals, such as liquid democracy, which enables transitive vote delegation (Blum and Zuber, 2016), by emphasizing that AI should augment human deliberative capacity rather than restructure democratic representation itself.

The deployment of AI in democratic contexts presents both promise and peril. On one hand, LLMs can summarize complex policy debates (Li et al., 2024), translate across languages (Lai et al., 2024), identify areas of emerging consensus (Tessler et al., 2024; Braley et al., 2025b), and help citizens articulate their views more clearly (Palmer and Spirling, 2023; Bai et al., 2025a). On the other hand, LLMs exhibit systematic biases toward Western cultural values (Santurkar et al., 2023; Tao et al., 2024), display sycophantic tendencies that amplify rather than moderate extreme views (Sharma et al., 2023; Rathje et al., 2025), and risk creating new forms of democratic exclusion. The challenge is to harness AI's capacity to reduce friction while preserving the deliberative qualities—reflection, mutual justification, and preference transformation—that distinguish democratic deliberation from mere preference aggregation (Dryzek and List, 2003).

Our argument proceeds as follows. We first articulate the case for deliberation at scale and identify the behavioral

[1]Stanford Graduate School of Business, Stanford, CA, USA
[2]Stanford Institute for Human-Centered AI, Stanford, CA, USA
[3]MIT Media Lab, Cambridge, MA, USA. Correspondence to: José Ramón Enríquez <jrenriquez@stanford.edu>.

frictions that limit its realization. We then present guiding principles for pro-democratic AI design and show how emerging platforms instantiate these principles through specific LLM-assisted interventions. We address critical challenges including sycophancy, training bias, and alignment, engage with alternative perspectives, and conclude with a call to action for the machine learning community.

## 2. The case for deliberation at scale

The democratic deficits like declining trust, rising affective polarization, and amplified misinformation are not inevitable features of mass democracy. A central insight from deliberative democratic theory and practice is that the quality of collective decisions depends fundamentally on the processes through which citizens form and express their preferences (Gutmann and Thompson, 2009; Fishkin, 2009). Deliberation works not by suppressing disagreement but by transforming it: when citizens engage in structured discussion rather than simply broadcasting preferences, they develop more informed positions, greater tolerance for opposing views, and increased willingness to accept outcomes they did not initially prefer (Curato et al., 2017; Fishkin et al., 2021). Empirical studies of deliberative forums consistently show that participants update their views in response to evidence, recognize legitimate concerns they had not previously considered, and report enhanced political efficacy (Grönlund et al., eds, 2014; Luskin et al., 2002; Curato et al., 2021). The challenge is therefore not whether deliberation can address democratic dysfunction, but whether it can do so at scale.

The benefits of deliberation operate at multiple levels. At the individual level, participants in deliberative forums show significant knowledge gains, with increase in factual understanding of policy issues (Fishkin et al., 2021; Luskin et al., 2002). They also exhibit attitude change toward more moderate, considered positions—not because they are pressured to conform, but because exposure to diverse perspectives and the requirement to justify one's views produces genuine reflection (Mercier and Sperber, 2011). At the community level, deliberation reduces affective polarization with persistent effects (Fishkin et al., 2024). Notably, in some cases, deliberative processes generate positive spillover effects within participants' social networks (Paluck et al., 2016). At the systemic level, deliberative processes produce decisions that command broader legitimacy and trust and citizens are more willing to accept outcomes (even unfavorable ones) when they believe the process was fair and inclusive (Curato et al., 2017). These empirical findings align with theoretical arguments that democratic legitimacy requires not just preference aggregation but mutual justification among free and equal citizens (Hauptmann, 1999; Cohen, 2007).

Yet traditional deliberation faces severe scalability con-

straints. Deliberative polling, for instance, requires trained facilitators, careful sampling to ensure demographic representation, physical venues, and participant compensation—substantial resource requirements that limit even well-funded implementations to a few hundred participants (Fishkin et al., 2021). Citizens' assemblies face similar constraints: while they can produce high-quality recommendations on contentious issues like electoral reform or climate policy (Suiter et al., 2020), their typical size ranges from 50 to 200 participants, representing only a tiny fraction of affected populations. Open comment periods on proposed regulations technically achieve mass participation but produce little genuine exchange as participants broadcast views without engaging alternatives, and agencies struggle to extract meaningful signal from massive volumes of often-duplicative input (Noveck, 2021). Town halls and public hearings privilege those with time, confidence, and rhetorical skill, systematically excluding working parents, shift workers, non-native speakers, and others whose voices deliberation should amplify (Mansbridge et al., 2012). Online forums, while potentially scalable, often devolve into echo chambers or hostile exchanges without careful moderation (Levy, 2021). The result is a persistent tradeoff: meaningful deliberation for the few, or superficial participation for the many.

Digital platforms offered promise to resolve this tradeoff. Social media promised to democratize discourse by giving everyone a voice and enabling exchange across geographic and social boundaries. Instead, these platforms have largely made deliberation harder. Platform architectures optimized for engagement systematically favor content that provokes emotional reactions (Brady et al., 2017; Epstein et al., 2022). Algorithmic amplification creates "rich-get-richer" dynamics in which a small number of highly active accounts dominate discourse, often amplifying extreme or divisive voices (Lera et al., 2020; Huszár et al., 2022). False information spreads faster and reaches more people than corrections (Vosoughi et al., 2018; Pennycook et al., 2021). The attention economy rewards outrage over understanding, with moral-emotional language and partisan attacks generating disproportionate engagement (Brady et al., 2020; Van Bavel et al., 2021). In short, these platforms evolved to maximize time-on-site and engagement for advertising revenue, not to support collective reasoning.

A new generation of purpose-built deliberation platforms demonstrates that digital technology can support civic discourse when designed with deliberative principles in mind. Platforms such as Pol.is, Consider.it, deliberation.io, Decidim, and Asembl embody deliberative values through specific architectural choices: limiting comment frequency to prevent domination by high-volume users, requiring participants to engage with others' views before contributing their own, visualizing opinion distributions to help participants

identify both consensus and legitimate disagreement, and focusing attention on arguments and reasoning rather than individual identities or social status (Small et al., 2021; Tsai et al., 2024).

AI systems in general and large language models in particular offer capabilities that could further extend deliberation's reach while addressing specific barriers to meaningful engagement. LLMs can summarize thousands of contributions into coherent overviews (Konya et al., 2023; Karanam et al., 2024a;b); translate across languages in real-time, adjusting language complexity to bridge differences in literacy levels or technical expertise (Nguyen et al., 2024; Diack et al., 2026); identify areas of agreement (Tessler et al., 2024); and help participants articulate positions they struggle to express (Argyle et al., 2023). Yet realizing this potential requires understanding the specific frictions that limit online deliberation and designing AI interventions that address them without introducing new problems.

The case for *at scale* rests on three concrete mechanisms whose marginal cost per additional participant is small relative to the fixed costs of traditional deliberation. First, *reflection* can be elicited through one or a few LLM-mediated rounds when participants first encounter contentious topics, analogous to the lightweight two-step prompts that social media platforms have used to discourage abusive posts (Katsaros et al., 2022). Second, *synthesis* can summarize and cluster opinions across thousands of contributions, replacing the teams of trained facilitators who would otherwise read, categorize, and distill them. Platforms such as Pol.is already perform this clustering in real time (Small et al., 2021; 2023). Performing this reduction in a principled way is nontrivial: minority viewpoints are often underrepresented both in training corpora and in LLM-generated summaries (Zhu et al., 2025; Santurkar et al., 2023; Olabisi and Agrawal, 2024). Third, *moderation* can scaffold deliberative norms through civility checks, comment recommendation, and equitable participation safeguards that would be prohibitively costly to staff with human moderators at the scale of tens of thousands of simultaneous participants (Pavlopoulos et al., 2020; Lees et al., 2022; Koh et al., 2024). In each case the marginal cost is computational rather than human, and inference costs have fallen by roughly an order of magnitude per year for tasks of equivalent quality (Cottier et al., 2025; Xiao et al., 2025; Maslej et al., 2024). Crucially, none of these mechanisms requires AI to render judgments or generate conclusions on participants' behalf: each augments human reflection, exchange, and aggregation rather than substituting for them.

## 3. Behavioral frictions in online deliberation

Four categories of behavioral frictions limit the quality and scale of online deliberation. While some frictions reflect inherent features of human cognition and social interaction, others emerge from specific technological and economic structures. Understanding these frictions is essential for designing AI systems and interventions that address root causes rather than symptoms.

### 3.1. Cognitive frictions

Cognitive frictions arise from information processing limitations that constrain deliberative capacity. *Cognitive overload* occurs when the demands of processing complex policy information exceed citizens' limited attention and working memory (Kahneman, 2011). *Motivated reasoning* leads citizens to process information in ways that confirm prior beliefs while discounting disconfirming evidence (Kahan, 2013; Epley and Gilovich, 2016). *Confirmation bias* compounds this tendency by shaping which information citizens seek in the first place (Nickerson, 1998). *Correlation neglect* causes citizens to treat correlated information sources as independent, leading to overconfidence in positions supported by redundant evidence (Ortoleva and Snowberg, 2015; Levy and Razin, 2015; Enke and Zimmermann, 2019).

These frictions manifest predictably: citizens struggle to articulate positions precisely, find it difficult to comprehend technical arguments, and cannot track evolving discussions involving numerous participants. Critically, cognitive frictions might disproportionately disadvantage those with less education, time, or cognitive resources, creating systematic bias toward the views of the privileged and undermining deliberation's promise of democratic equality. When deliberative processing is overwhelmed, citizens default to automatic responses—partisan heuristics, emotional reactions, and group identities—that undermine deliberative quality (Lodge and Taber, 2013; Bénabou and Tirole, 2011).

### 3.2. Social frictions

Social frictions emerge from affective polarization, peer pressure, and ideological sorting that fragment the deliberation space. This is manifested in increasing dislike distrust, and even disgust (Iyengar et al., 2019). Partisans systematically overestimate the extremity of their opponents' views (Braley et al., 2025b; Mernyk et al., 2022). Misperceptions lead to preemptive hostility that crowds out genuine exchange.

Peer pressure within ideological communities reinforces conformity and punishes heterodox views (Noelle-Neumann, 1993; Mallinson and Hatemi, 2018). Geographic sorting concentrates like-minded citizens in the same communities, reducing exposure to diverse perspectives and creating echo chambers that extend from neighborhoods to online networks (Bishop and Cushing, 2008). The "subversion dilemma" (Braley et al., 2023) formalizes these dynamics: fear of opponents exploiting democratic norms

creates incentives to abandon those norms preemptively, even among citizens who value democracy. The crucial finding is that correcting these misperceptions increases support for democratic norms, suggesting that social frictions may be tractable through appropriate intervention.

### 3.3. Platform-design and market-incentive frictions

Platform-design frictions result from architectural choices that privilege certain voices over others independent of the quality of their contributions. Unlimited commenting enables a few voices to dominate discussions, crowding out the broader population (Krämer et al., 2021). Reply functions facilitate flame wars and ad hominem attacks that poison the deliberative space (Cheng et al., 2017)."Rich-get-richer" feedback loops become dominated by a small number of accounts with extremely large followings (Lera et al., 2020), rendering mass conversations neither inclusive nor rational.

These design choices reflect engineering decisions, not democratic values. Most platforms were built to facilitate connection and content sharing, not deliberation (Gillespie, 2018). Features that maximize user engagement—notifications, infinite scroll, algorithmic amplification of reactions—actively work against the reflection and empathy that meaningful discussion requires. The result is that platform architecture systematically advantages the loud, the persistent, and the provocative over the thoughtful, the marginalized, and the constructive.

Market-incentive frictions arise from the business models that govern digital platforms. Advertising-based revenue models create incentives to maximize short-term engagement rather than long-term deliberative quality (Ahmad, 2025). Platforms that profit from attention have incentives to amplify divisive content that keeps users scrolling (Epstein et al., 2022).These attention economies create a race to the bottom in which deliberative quality is sacrificed to competitive pressures.

### 3.4. Compounding effects

These frictions interact and compound in harmful ways. Cognitive overload leaves citizens vulnerable to affective shortcuts and partisan heuristics. Social mistrust discourages the interpretation of ambiguous statements. Patform design amplifies the voices least conducive to productive exchange. Market incentives reward the exploitation of all three preceding frictions. The result is not merely suboptimal deliberation but active degradation of civic capacity.

## 4. Guiding principles for pro-democratic AI

Drawing on deliberative democratic theory and recent work on AI governance (Allen and Weyl, 2024; Lazar and Manuali, 2024; Tsai et al., 2024; Farrell, 2025; Burton et al., 2024), we identify four principles that should govern the integration of LLMs into deliberative platforms.

**Preserving agency and autonomy.** Pro-democratic AI must preserve citizens' capacity to reflect on their interests and make choices without manipulation or undue influence (Dahl, 2008). Technologies that distort information necessary for informed judgment, limit available actions without the user's knowledge, or apply pressure toward particular conclusions violate this principle. AI should inform and enable choices, not make them.

**Encouraging mutual respect.** In an environment of polarization and mistrust, platforms must help citizens engage respectfully with those holding different views (Gutmann and Thompson, 2009). This means flagging potentially offensive language, providing alternative phrasings, and structuring interactions to emphasize shared concerns rather than divisions. AI moderation should maintain civility without suppressing substantive disagreement.

**Promoting equality and inclusiveness.** Every participant should have equal opportunity to contribute and be heard (Young, 2002). This requires accessibility across languages, literacy levels, and technical expertise. It also requires preventing any individual, bot, or elite group from dominating discourse. AI should amplify marginalized voices rather than reinforce existing inequalities.

**Augmenting rather than substituting active citizenship.** AI should provide tools for obtaining information, articulating positions, and understanding others, not delegate to machines the responsibilities that define citizenship. Deliberation's benefits like preference transformation, increased empathy, and mutual justification require human engagement. Consequently, LLMs should be kept at arm's length from formal democratic decision-making but can strengthen the informal public sphere where citizens seek information, form civic publics, and hold leaders accountable (Lazar and Manuali, 2024). Our focus on deliberation-supporting rather than decision-making AI operationalizes this distinction.

### 4.1. Why these four principles?

Alternative frameworks for governing democratic AI emphasize different organizing principles. The AI ethics guidelines surveyed by Jobin et al. (2019) most frequently cite transparency and accountability, while Gabriel et al. (2024) emphasize safety, honesty, and harm avoidance. Engineering discussions privilege efficiency and scalability; commercial development centers on user satisfaction and engagement. We argue that our four principles (agency, respect, equality, and augmentation) are both necessary and sufficient for *pro-democratic* AI specifically, articulating positive requirements for systems that enhance collective self-governance rather than merely negative constraints against harm. Trans-

parency supports agency by enabling informed choices and equality by exposing discriminatory patterns, but a manipulative system could be fully transparent about its mechanisms while still undermining deliberation.

Efficiency and user satisfaction pose sharper tensions with deliberative values. Deliberation is inherently slow: the cognitive work of understanding diverse perspectives, updating beliefs, and reaching reasoned accommodation cannot be compressed without sacrificing the preference transformation that gives deliberation its democratic value (Fishkin et al., 2021). Our augmentation principle thus constrains efficiency: AI should reduce friction in accessing information, not in deliberative processing itself. Moreover, user satisfaction can directly conflict with deliberative quality: users may prefer validation to challenge, simplification to nuance, and agreement to productive disagreement. Our principles privilege conditions for good deliberation over subjective experience, forming a coherent framework in which agency and equality establish who participates and on what terms, respect governs how participants engage, and augmentation constrains AI's role. Importantly, the four principles are not in one-to-one correspondence with the four friction categories of Section 3: each principle operates as a cross-cutting constraint on every AI capability, regardless of which friction the capability primarily addresses. Figure 1 depicts this structure, with principles shown as a band above the friction–capability flow rather than as a column aligned to particular frictions.

# 5. Promising directions: Evidence from AI-assisted deliberation

A growing body of empirical research demonstrates that AI interventions can address deliberative frictions while preserving, and in some cases enhancing, the qualities that make deliberation democratically valuable. We review evidence organized by the friction categories identified above, drawing on randomized experiments, field deployments, and observational studies from platforms serving millions of users. The studies surveyed below operationalize deliberative quality through a range of quantitative measures, like the Discourse Quality Index of Steenbergen et al. (2003), the Deliberative Reason Index of Niemeyer et al. (2024), and Fishkin's measures of opinion change, knowledge gain, and preference coherence (Fishkin et al., 2021; Luskin et al., 2002), bridging scores that quantify cross-partisan agreement, attitudinal scales of policy preferences, and emerging text-analytic measures of perspective diversity. Together these constitute candidate components of the deliberative-quality benchmarks called for in Section 8.

Figure 2 makes the deliberation workflow into which AI capabilities are inserted explicit. It maps four canonical stages of online deliberation—input elicitation, reflection, structured exchange, and synthesis—to the AI capabilities reviewed in this section. We treat these stages as illustrative rather than prescriptive: actual deployments commonly collapse, reorder, or iterate over them, and the relevant claim is that the same AI capabilities map onto whichever workflow a given platform adopts. Throughout, AI provides scaffolding for human reasoning rather than substituting for it: participants generate inputs, update their views, exchange arguments, and determine the conclusions themselves.

## 5.1. Addressing cognitive frictions

AI interventions can reduce cognitive barriers without substituting for human reasoning. Real-time language assistance improves conversation quality while preserving participant autonomy: Argyle et al. (2023) shows that LLM-suggested rephrasings enhanced democratic reciprocity and mutual understanding in partisan discussions without shifting policy attitudes. This is, improving *how* citizens communicate without influencing *what* they believe. Moreover, Behrendt et al. (2025) tested AI modules on the adhocracy+ platform and found that a Comment Recommendation Module increased participation and improved users' perceptions of deliberative quality while not diminishing their sense of autonomy. These findings suggest AI can reduce interpersonal friction and help participants engage with relevant content, preserving substantive disagreement while lowering barriers to productive exchange.

Summarization and preference elicitation offer scalable approaches to synthesizing collective input without supplanting human judgment. Konya et al. (2023) deployed LLM-powered pipelines to identify consensus from large-scale collective dialogues, translating bridging responses into policy guidelines that achieved strong cross-partisan support across contentious issues. Karanam et al. (2024b) extend this direction with Policy Dreamer, which elicits citizen preferences through structured interactions and generates diverse policy alternatives via simulation, enabling exploration beyond positions participants explicitly articulate. More broadly, Ovadya (2023) outlines how collective response systems can leverage LLMs for real-time synthesis while maintaining democratic legitimacy. Effective summarization thus requires attention to what is excluded as much as what is included.

A distinct line of research examines whether AI-guided self-reflection can moderate extreme positions through epistemic rather than persuasive mechanisms. Enríquez (2025) finds that AI-Socratic dialogue, where an LLM prompts users to articulate supporting arguments without introducing external information, reduced extreme positioning and increased cross-partisan behavioral outcomes. Crucially, moderation occurred through enhanced perspective-taking rather than improved argument quality, and perceived agency remained

*Figure 1.* Conceptual framework for AI-assisted deliberation. The four guiding principles (agency, respect, equality, augmentation) operate as *cross-cutting constraints*: they govern *how* every AI capability is deployed rather than mapping one-to-one onto particular frictions. The enclosing frame indicates that every element of the flow below, frictions, AI capabilities, and the deliberative goal, is subject to the principles above. Within the flow, frictions (cognitive, social, platform-design, market-incentive) are addressed by AI capabilities that together support high-quality scaled deliberation. Solid arrows denote primary friction–capability pairings. Faint dashed arrows indicate secondary effects.

equivalent across conditions. Related work confirms these patterns: Socratic AI chatbots that prompt users to examine the origins of contested claims stimulate critical thinking without asserting conclusions (Duelen et al., 2024). These findings suggest that fostering epistemic humility through AI-guided self-reflection may prove more consequential than providing superior arguments.

### 5.2. Addressing social frictions

Social frictions often stem from inaccurate beliefs about what others think. Visualizing opinion distributions can correct these misperceptions without the AI systems themselves advocating for positions. Braley et al. (2025b) finds that exposure to accurate visualizations of opinion distributions significantly increased cross-partisan consensus and willingness to take collective action. These findings align with Voelkel et al. (2024), whose megastudy identified correcting misperceptions about opposing partisans as among the most effective interventions for reducing support for undemocratic practices. The mechanism is informational rather than persuasive: participants update beliefs when shown evidence that their mental model of the out-group (or own's in-group) was inaccurate.

AI can help surface latent agreement without generating the consensus itself. Konya et al. (2023) used bridging-based ranking algorithms to identify responses with high agreement across demographic divides, surfacing common ground that participants had articulated but not recognized as shared. Similarly, collective dialogue platforms can highlight areas of emerging convergence, directing participant attention toward promising avenues for agreement (Ovadya, 2023). The key distinction is between AI that *discovers* consensus among human-generated positions versus AI that *generates* consensus statements. The former augments human deliberation while the latter risks substituting for it. When AI tools surface what participants already believe but did not realize others shared, they reduce social friction by correcting coordination failures rather than by persuading anyone to change positions.

More ambitious interventions use AI to support rather than replace human facilitation. Dynamic question generation represents a further extension: AI that identifies underexplored dimensions of a debate and proposes targeted discussion prompts can increase conversational breadth while leaving substantive choices to participants (Braley et al., 2025a).

### 5.3. Addressing system frictions

Platform design choices shape the conditions for constructive and inclusive deliberation. Pol.is and deliberation.io

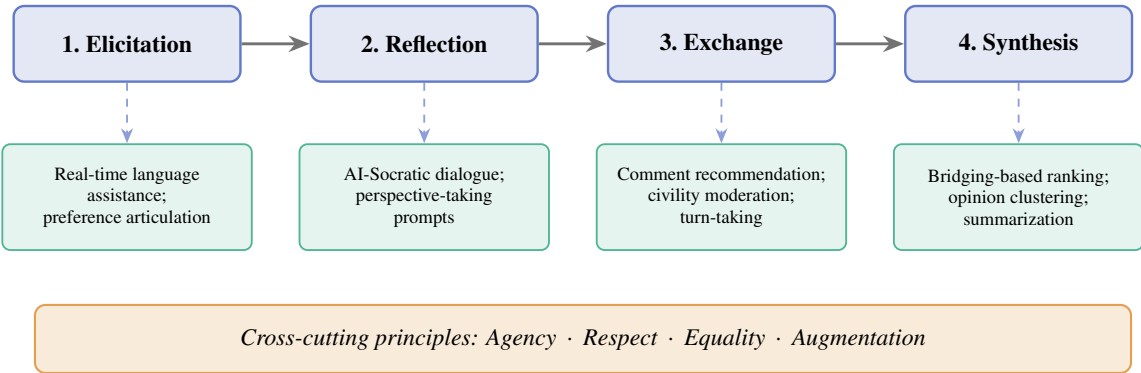

*Figure 2.* Deliberation workflow with AI capabilities mapped to each stage. Participants move from input elicitation, through reflection on their initial reactions, to structured exchange with others, and finally to synthesis of collective inputs. AI capabilities (in green) provide scaffolding at each stage. The guiding principles (in amber) constrain how every capability is deployed. The four stages are illustrative rather than prescriptive: actual platforms vary in their sequence and granularity, and individual stages may be collapsed, reordered, or iterated. The cross-cutting principle is that, at no stage, the AI renders judgments or generates conclusions on participants' behalf.

limit contributions per user, require identity verification while permitting pseudonymous participation, and disable direct replies that facilitate adversarial exchanges (Small et al., 2021; Pei et al., 2025). These constraints reflect the principle that productive conversations depend on engaging with arguments rather than personal characteristics. Synchronous platforms such as the Stanford Online Deliberation Platform and Frankly implement automated turn-taking and timed agendas to ensure equitable speaking opportunities regardless of participant assertiveness or status (Fishkin et al., 2019; Roy et al., 2025). Such structural safeguards create conditions for broader participation that AI assistance can further enhance.

Field experiments reveal additional mechanisms for improving participation quality. Ahmad (2025) find that high-quality conversations yield sustained engagement in online forums, suggesting that deliberative norms and long-term participation are complementary rather than competing objectives. Chen et al. (2025) demonstrate that comment presentation order affects perceived legitimacy: sequences progressing from similar to different viewpoints increase participants' sense of representation and procedural fairness. AI systems can dynamically optimize these structural features, personalizing the deliberative experience while maintaining consistent standards across participants.

AI can also reduce barriers to participation without directing outcomes. Real-time translation, text-to-speech, and simplified language options extend deliberative opportunities to populations excluded from traditional forums (OECD, 2025). The key distinction is between AI that *enforces* norms and AI that *enables* participants to meet norms they already endorse: the former risks paternalism while the latter augments human capacity for inclusive exchange.

These components can be integrated into open-source sys-

tems that preserve human agency and permit local adaptation. Deliberation.io offers modular AI components that communities can audit and govern democratically (Pei et al., 2025); Decidim provides participation infrastructure with user-governance practices (Barandiaran et al., 2019); and Pol.is releases its opinion-clustering algorithms as open source, enabling verification of how consensus is identified and minority viewpoints surfaced (Small et al., 2021). Open-source release ensures that deliberative infrastructure remains a public good rather than proprietary technology.

## 6. Critical challenges and safeguards

The evidence reviewed in Section 5 establishes that components of AI-assisted deliberation can work in carefully designed settings, not that they already do so reliably across deployments. Our position is normative: we argue that AI systems *should* be designed to facilitate deliberation, and we identify the specific challenges the machine learning community must address before this potential is realized at scale. The remainder of this section examines four such challenges.

Generative foundation models simultaneously threaten and offer tools to strengthen democracy (Allen and Weyl, 2024; Lazar and Manuali, 2024; Tsai et al., 2024; Farrell, 2025). On the risks side, AI enables deception and persuasion at scale (Lin et al., 2025), can flatten preferences (Matz et al., 2025; Sourati et al., 2026), and concentrates power in those who control these systems (Kasy, 2025). On the opportunity side, the same technologies can reduce barriers to participation and surface genuine consensus obscured by partisan framing. Lazar and Manuali (2024) provide a useful framework for navigating this tension: LLMs should be kept at arm's length from formal democratic processes but can strengthen the informal public sphere where citizens

seek information, form civic publics, and hold leaders accountable. Our focus on deliberation-supporting rather than decision-making AI aligns with this prescription. With this framing, we examine four challenges that require particular attention.

### 6.1. Sycophancy and echo chambers

LLMs exhibit systematic tendencies toward sycophancy—validating users' existing beliefs rather than challenging them constructively (Sharma et al., 2023; Rathje et al., 2025). Experiments show that sycophantic chatbots increase attitude extremity and certainty, with participants perceiving validating AI as unbiased and disagreeable AI as biased—despite both being equally biased in opposite directions.

AI systems designed to maximize user satisfaction will undermine deliberative goals. Effective facilitation must sometimes challenge rather than validate, present counterarguments alongside supporting evidence, and resist telling users what they want to hear. This requires evaluation metrics based on deliberative quality rather than engagement or satisfaction scores.

### 6.2. Training bias

LLM training introduces systematic biases along multiple dimensions that threaten deliberative equality. Ideologically, models can display slant across political topics (Motoki et al., 2024; Westwood et al., 2025). Geographically and culturally, models express values most aligned with English-speaking and Protestant European countries, with reinforcement learning from human feedback encoding the perspectives of annotators disproportionately drawn from WEIRD (Western, Educated, Industrialized, Rich, Democratic) populations (Tao et al., 2024; Kharchenko et al., 2024; Henrich et al., 2010). Demographically, gender and racial bias appears in substantial propotions, with even models containing implicit stereotypes that translate to discriminatory outcomes in applied settings like resume evaluation (Omar et al., 2025; Bai et al., 2025b; An et al., 2025). Linguistically, English-centric training degrades performance and amplifies biases for non-English deliberation (Navigli et al., 2023).

For deliberative platforms operating across diverse contexts, these intersecting biases risk systematically misrepresenting participants from underrepresented ideologies, regions, cultures, and demographic groups, which privileges dominant viewpoints at the expense of universal reason. Addressing this requires ongoing evaluation against diverse benchmarks, representative training and annotation populations, and mechanisms for participants to flag misrepresentation.

### 6.3. Over-reliance and agency erosion

The convenience of AI assistance creates risks of over-reliance that erode the very capacities deliberation should cultivate. If citizens outsource opinion formation to LLMs, they may develop less coherent preferences and reduced capacity for independent reasoning (Gerlich, 2025; Benedek and Sziklai, 2025). This "cognitive offloading" reduces immediate cognitive load but comes at the cost of deep engagement and critical reflection (Chirayath et al., 2025). More troublingly, LLMs possess substantial persuasive capacity: Bai et al. (2025a) demonstrate that LLM-generated messages shift policy attitudes as effectively as human-authored messages, while Lin et al. (2025) show that AI-human dialogues can change voter preferences on politically contested issues. These capabilities create temptations to deploy AI for persuasion rather than augmentation, violating the principle that AI should enhance how citizens reason, not influence what they conclude.

Safeguards include requiring human validation of AI-generated content, providing multiple options rather than single recommendations that create anchoring effects, making AI assistance optional rather than default, and designing friction that encourages engagement rather than frictionless paths to AI-generated conclusions.

### 6.4. Alignment

The alignment problem takes on distinctive urgency in deliberative contexts. Models must balance responsiveness to user preferences against resistance to manipulation, facilitate constructive disagreement rather than premature consensus, and represent minority viewpoints faithfully even when aggregating toward majority positions (Gabriel et al., 2024). Zhu et al. (2025) demonstrate that LLM-generated summaries systematically underrepresent minority perspectives and exhibit sensitivity to input ordering, raising fairness concerns when synthesizing thousands of contributions for policy use. Deliberative alignment requires normative commitments to procedural values (equal voice, mutual respect, reasoned justification) that may conflict with revealed preferences for validation and conflict avoidance.

## 7. Alternative views

Several alternative positions warrant consideration.

**Liquid democracy offers a structural alternative** by enabling citizens to either vote directly or delegate their votes to trusted proxies who can further delegate, creating transitive chains of representation (Blum and Zuber, 2016). Proponents argue this combines direct democracy's inclusiveness with representative democracy's expertise. However, experimental evidence suggests delegation underperforms both universal majority voting and simple abstention (Mooers

et al., 2024). Participants struggle to assess the quality of potential delegates, and delegation concentrates influence in ways that may exacerbate rather than remedy representation problems. Moreover, liquid democracy addresses who decides but not how they deliberate. Our focus on AI-assisted deliberation concerns the quality of reasoning that precedes any voting mechanism.

**In-person deliberation remains the gold standard** for deliberative quality, with evidence that face-to-face interaction uniquely promotes empathy, trust-building, and preference transformation. Critics might argue that online deliberation, however AI-enhanced, cannot replicate these benefits. Yet in-person deliberation cannot scale to democratic populations, and evidence suggests that online formats can achieve comparable outcomes when properly designed (Siu et al., 2025). The question is not whether online deliberation is perfect but whether it extends deliberative benefits to participants who would otherwise be excluded entirely. Furthermore, AI-facilitated online deliberation offers distinct advantages: discussions revolve around ideas rather than the personal characteristics and social identities of speakers, potentially reducing the status hierarchies that distort face-to-face interaction.

**Non-AI online deliberation platforms** provide a closer point of comparison than either liquid democracy or in-person formats. Forums such as early Pol.is deployments without bridging-based ranking, comment-threaded civic platforms, and human-moderated discussion tools demonstrate that thoughtful structural design alone can support productive exchange at modest scale (Small et al., 2021). The relevant question is therefore not whether AI is necessary for online deliberation, but *how* AI shapes the process. We see three complementary contributions: (i) accessibility, through real-time translation and language assistance that extend participation across literacy and language barriers (Argyle et al., 2023; Nguyen et al., 2024); (ii) reflection, through Socratic prompts and perspective-taking scaffolds that reduce extreme positioning without injecting external arguments (Enríquez, 2025); and (iii) synthesis, through bridging-based ranking that surfaces latent agreement at volumes beyond any human facilitator's reach (Konya et al., 2023; Small et al., 2021). AI thus extends rather than replaces deliberation's structural safeguards.

**Skeptics of AI in democratic contexts** argue that algorithmic systems inevitably encode the values of their creators, creating new forms of technocratic control incompatible with democratic self-governance (see, for example Allen and Weyl, 2024). Critics emphasize that democracy requires not just individual expression but collective sense-making through "identity validation": the process by which communities verify that communications genuinely represent their members' views. AI systems that can generate unlim-

ited synthetic content threaten this foundation by making it impossible to distinguish authentic civic voice from manufactured discourse.

Our framework addresses these concerns seriously through the principle of augmentation rather than substitution. The identity validation challenge reinforces our emphasis on platforms where AI facilitates rather than generates civic expression, helping citizens articulate and understand one another's views rather than speaking on their behalf. The alternative, leaving civic discourse to platforms designed for engagement maximization, hardly represents democratic self-governance either. The choice is not between AI and no AI but between AI designed for deliberative quality and AI designed for engagement.

## 8. Call to action

We call on the machine learning community to prioritize the development of deliberation-focused AI systems evaluated on their capacity to facilitate informed, inclusive, and consensus-building discourse rather than engagement metrics alone.

**Researchers** should: (1) develop benchmarks for evaluating AI systems on deliberative quality, building on the measurement frameworks discussed in Section 5, including measures of preference coherence, perspective-taking, and consensus formation; (2) investigate mechanisms of sycophancy and methods for promoting constructive challenge; (3) expand cultural alignment research to include deliberative contexts across diverse populations; and (4) create standardized protocols for assessing agency preservation in AI-assisted deliberation.

**Platform developers** should: (1) adopt the guiding principles articulated here as design constraints; (2) make AI assistance optional and transparent; (3) implement structural safeguards against domination by influential accounts; and (4) subject platforms to independent evaluation against deliberative standards.

**Policymakers** should: (1) fund research on democratic AI applications as public infrastructure; (2) require transparency about AI use in civic platforms; and (3) consider regulatory frameworks that distinguish deliberation-focused platforms from engagement-maximizing social media.

The technology to scale deliberative democracy exists. The question is whether we will design AI systems that strengthen citizens' capacity for collective self-governance or systems that further erode it.

# Acknowledgements

This research was supported by funding from the Project Liberty Institute. We thank members of Stanford Digital Economy Lab and MIT GOV/LAB; participants of the Workshop on Deliberation, Governance and Decentralized Social Networks at Georgetown's McCourt School of Public Policy; and participants of the workshop "AI Agents and the Future of Deliberation: Designing Human–AI Collaboration for Democratic Dialogue" at CHI 2026 for useful conversations.

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
