# OpenReview forum: "Position: AI Should Facilitate Democratic Deliberation at Scale"
_ICML.cc/2026/Position_Paper_Track — ICML 2026 Position Paper Track spotlight_

### Official Review · Reviewer_H3Fn · 2026-03-04

**Significance:** 4
**Argument Clarity:** 4
**Rating:** 6
**Confidence:** 4

**Questions:**

None, this is an exceptionally clear paper.

**Alternative Views Section:**

Yes

**Compliance With Llm Reviewing Policy A Conservative:**

Affirmed.

**Discussion Potential:**

3

**Final Justification:**

I had few concerns about this paper and continue to think it should be accepted.

**Paper Summary:**

The authors make the case for AI-assisted democratic deliberation. They survey the benefits of deliberation on increasing mutual respect and understanding and in leading to group decisions that are seen as more legitimate and trustworthy. Then, the authors highlight frictions (like cognitive biases and social pressures) to deliberation in online settings (which scales better than in-person deliberation, which poses logistical challenges). Based on these frictions, they propose guidelines for developing AI deliberation systems and highlight ways in which such systems could address different frictions. The paper concludes with challenges in developing such systems, alternate views, and a call to action outlining how researchers, developer, and policymakers can contribute to developing successful AI-assisted deliberative tools.

**Position:**

Yes

**Position In Title:**

Yes

**Related Work:**

4

**Strengths And Weaknesses:**

### Strengths
- The structure and writing of the paper are excellent. It is clear and build towards a central thesis.
- Claims and proposals are well-supported by evidence
- The authors do an excellent job of highlighting challenges in the use of AI
- The call to action is clear and specific
- The topic is important and interesting to the ML community, highlighting a societally impactful use of the technology with thoughtful psychological and political science context

### Weaknesses
- As a position paper, there is little to find fault with. The only thing that raised an eyebrow was the highlighting of liquid democracy as the main contrasting approach (called out in the abstract and as the first alternative view). Liquid democracy, as it is a voting system, seems quite different from AI-assisted deliberation. One could imagine using both. Other approaches to deliberation seem like the more relevant contrast. For instance, non-AI online deliberation/discussion systems, or various in-person formats like caucuses.

### Minor
- Line 25 (right col) "Mansbridge’s" cite?

**Support:**

4

---

> ### Author Rebuttal · Authors · 2026-03-30
>
> We are grateful for Reviewer H3Fn's careful assessment.
>
> **Liquid democracy as a contrast.**
> Our choice to contrast with liquid democracy is because that model is the most prominent technology-enabled model for citizen participation. We think the contrast is productive because AI-facilitated liquid democracy (where an LLM selects delegates or optimizes delegation chains on behalf of users) would concentrate decision-making authority in the model rather than augmenting citizens' own reasoning. This is precisely the kind of substitution we warn against. AI-assisted deliberation, by contrast, keeps citizens as the reasoners and decision-makers throughout.
>
> That said, the Reviewer is right that liquid democracy and AI-assisted deliberation are not mutually exclusive. One could envision hybrid systems where AI-facilitated deliberation informs the reasoning that precedes a liquid democracy vote. Our argument is that focusing on improving *how* citizens reason avoids the risk of outsourcing decisions to AI systems, which is inherent in restructuring *who* decides.
>
> Regarding non-AI deliberation systems: our second alternative view already engages with in-person deliberation as the gold standard and emphasizes its non-scalable nature (e.g., deliberative polls limited to a few hundred participants, citizens' assemblies to 50-200). In the next version of the paper, we will elaborate further on this contrast and acknowledge non-AI online deliberation platforms as an additional comparison point, clarifying what AI specifically adds.
>
> **Minor: Mansbridge citation.**
> Thank you very much for spotting the typo. We will complete the reference to read: "This position draws on a rich tradition of democratic theory --- from Habermas's theory of communicative action (1984) through Landemore's open democracy (2020) to Mansbridge's (2012) --- which holds that legitimate collective decisions emerge through reasoned exchange among free and equal citizens."
>
> We thank the Reviewer again for the generous assessment. If there are any further points we can address, we would be happy to do so.

---

> > ### Author Rebuttal · Reviewer_H3Fn · 2026-04-02
> >
> > Thank you for the clarification about liquid democracy. The suggested update sounds good.

---

### Official Review · Reviewer_nBNV · 2026-03-12

**Significance:** 4
**Argument Clarity:** 4
**Rating:** 6
**Confidence:** 3

**Questions:**

- Do the authors believe this is something that can be realized organically from the community as a whole, or would this require legislation? I ask because legislation is slow, legislators perhaps have an incentive to avoid deliberation if they are currently on the right side, and we currently live in a work where diversity, inclusion, and equity (three words that ought to be considered positive) are frowned upon.
- Related to the question above, if this is not to be achievable through legislation, do the authors believe that the current stakeholders would be willing to change their business practices? I might be biased by my news sources, but I hear of shady practices to actively encourage engagement (which is touched upon in the paper). If that is the revenue. source, why would they harm?
- If an LLM is effectively a mediator of deliberation at scale, how do we ensure that biases baked into the model do not influence the process? There is plenty of evidence that safe guards on current LLMs only ensure shallow bias mitigation.

**Alternative Views Section:**

Yes

**Compliance With Llm Reviewing Policy A Conservative:**

Affirmed.

**Discussion Potential:**

4

**Paper Summary:**

The authors in this paper argue that AI should be used to overcome many of the tensions that exist today (partly due to the use of AI and especially in social media) around democratic deliberation. In particular the authors argue that the use of AI allows democratic deliberation at scale so that everyone should have an equal voice and that everyone should have agency in the democratic process. The topic is important, it is especially relevant in the current (almost worldwide) political environment(s), and the paper is well thought out and the points expressed in a balanced and clear way.

**Position:**

Yes

**Position In Title:**

Yes

**Related Work:**

3

**Strengths And Weaknesses:**

**Strengths**
- The authors do a good job highlighting the issues, and break them down nicely into functional cognitive, social, and system issues.
- The four guiding principles are nicely laid out, and then justified.
- It is hard to argue that the position put forward is not timely. Sadly, one might conclude the call for action is a little too late. Perhaps if we can foster discussion and build infrastructure to better support democratic deliberation at scale, then one might hope that we can heal the divisions that currently exist.
- The arguments are put forward in a way that is balanced. AI can be used as a solution, but there are risks and caveats, which themselves are considered.

**Weaknesses**
- In the paper the authors discuss a number of platforms that exist for deliberation — is the proposal to move more in that direction, or a different direction? It is not clear to me how much this has already been attempted as a solution?
- There is an issue of “power” asymmetry in that democratic deliberation is facilitated by AI. The users are interacting with AI that is development by a specific entity — that entity essentially has control.
- The owners of the online echo chambers have a vested interest in maintain the echo chambers. I appreciate that this is called out in the paper, but it will clearly be a barrier.

**Support:**

4

---

> ### Author Rebuttal · Authors · 2026-03-30
>
> We thank Reviewer nBNV for the assessment and for the relevant questions about feasibility and implementation.
>
> **Novelty relative to existing platforms.**
> Our proposal seeks to both extend existing digital platform efforts and to suggest new paths, since the work by these has been insufficient. Platforms like Pol.is (vTaiwan), Decidim, and deliberation.io are early instantiations of deliberative principles in digital infrastructure, from which our paper draws on (see Section 5). While some platforms have been deployed locally, most of design decisions either follow heuristics or optimize for local conditions, rather than follow guided principles. For example, Pol.is uses bridging-based ranking algorithms to surface cross-partisan agreement (Konya et al., 2023), but does not incorporate AI-guided reflection at the input stage; deliberation.io deploys Socratic dialogue to prompt users to examine their own reasoning (Enríquez, 2025; Pei et al., 2025), but does not yet integrate automated opinion synthesis; and the Stanford Online Deliberation Platform implements automated turn-taking for equitable participation (Fishkin, 2019), but relies on human facilitators for content moderation. Each platform addresses a subset of frictions with a subset of AI capabilities. That has direct implications in what type of AI-human interactions we can and *should* scale. In that sense, our main contribution is to provide suggestive directions to what is currently missing: a principled framework for how the ML community should approach the design of AI-assisted deliberation systems and specific areas of improvement (e.g., metrics; AI scaffolding rather than substitution)
>
> **Power asymmetry, bias, and control.**
> The Reviewer raises a foundational concern: whoever controls the AI (and more specifically, whoever determines what AI maximizes; see Kasy, 2025) holds disproportionate power. We agree this is important, and both concerns, power concentration and LLM bias, have a common answer: the principle of agency and autonomy. If AI assistance is transparent, optional, and users make the final decisions, then neither the entity controlling the model nor the biases embedded in it can determine deliberative outcomes. Our framework enforces this by constraining AI to scaffolding roles: AI assists humans to reflect and identify areas of collaboration, but does not render judgments or generate conclusions on their behalf. As long as humans remain the arbiters, power over the AI does not translate into power over the deliberation and/or outcomes. This principle is reinforced by our emphasis on open-source infrastructure, enabling independent audits, community governance, and local adaptation.
>
> **Organic adoption vs. legislation.**
> We want to be clear that our proposal does not depend on legislation. Purpose-built deliberation platforms can be adopted organically by civic organizations, academic institutions, and local governments (several already operate on this model). This can be facilitated through open-source and modular platforms.
>
> However, regulation can definitely create enabling conditions that facilitate constructive deliberative environments. This can be complemented by work at academic institutions and civic organizations, who can set regulatory technical specifications and benchmarks. In parallel, we argue that for-profit organizations might maximize long-term, sustained engagement by promoting high-quality interactions over viral engagement models.
>
> We are grateful for the Reviewer's engagement with these implementation questions and welcome any further suggestions on how to strengthen this discussion.

---

> > ### Author Rebuttal · Reviewer_nBNV · 2026-04-03
> >
> > Thanks for the response. I will maintain my positive score of the paper.

---

### Official Review · Reviewer_okNN · 2026-03-13

**Significance:** 3
**Argument Clarity:** 2
**Rating:** 4
**Confidence:** 3

**Questions:**

see weaknesses

**Alternative Views Section:**

Yes

**Compliance With Llm Reviewing Policy A Conservative:**

Affirmed.

**Discussion Potential:**

3

**Final Justification:**

The authors' response has addressed most of my previously raised concerns, and I would like to increase my overall rating to a 4.

**Paper Summary:**

This paper examines the concept of using LLMs to facilitate democratic deliberation at scale. The authors argue that AI systems should be designed to facilitate democratic deliberation rather than replace human judgment or restructure democratic representation. They identify four major categories of barriers to effective democratic deliberation: cognitive, social, platform-design, and market-incentive frictions. It then proposes a framework for AI-assisted deliberation based on four guiding principles: agency, respect, equality, and augmentation.

According to the authors, AI tools can support democratic discourse through capabilities such as summarization, translation, consensus discovery, and facilitation, while also discussing risks such as sycophancy, training bias, over-reliance, and alignment. In addition, it concludes with a call for the ML community to build and evaluate deliberation-focused AI systems using metrics tied to deliberative quality rather than engagement.

**Position:**

Yes

**Position In Title:**

Yes

**Related Work:**

3

**Strengths And Weaknesses:**

## Strengths
- The paper addresses a timely and socially important question: how generative AI systems might shape democratic discourse and civic participation. The core position and arguments are well-motivated, explicit, and easy to follow. The paper is well organized and form a coherent conceptual framework.
- The paper does not adopt a purely optimistic perspective. It also carefully discusses key risks, which balances treatment improves the credibility of the proposal.
- For related work, the paper cites both deliberative democracy sources and recent AI-related studies, making its contextual grounding complete.

## Weaknesses
- The paper advocates evaluating AI systems based on "deliberative quality" rather than engagement metrics. However, it does not sufficiently operationalize what measurable indicators of deliberative quality would look like. For example, how should perspective-taking be quantified and how should consensus quality be evaluated?
- The authors assume that AI systems can reliably perform tasks such as consensus identification, fair summarization, and unbiased facilitation. However, in practice, LLMs may introduce framing bias, summarization bias, or subtle ideological distortions. The authors may overestimate the capabilities of AI models, which should be better discussed in the paper.
- The paper argues that AI-assisted deliberation is a promising path for enabling democratic deliberation at scale. However, most of the examples discussed in Sec. 5 focus on relatively localized interventions, such as rephrasing assistance, comment recommendation, and consensus surfacing. While these tools may improve specific aspects of online discussions, they provide limited evidence that AI can serve as a central mechanism for large-scale democratic deliberation. As a result, the scope of the paper’s central claim appears to exceed the strength of the supporting evidence.

**Support:**

2

---

> ### Author Rebuttal · Authors · 2026-03-30
>
> We thank Reviewer okNN for the thoughtful engagement, for recognizing the timeliness and contribution of our paper, as well as for pushing for metrics, discussion of AI capabilities, and to describe the scope of the deliberation.
>
> **Deliberative quality metrics.**
> We agree with the Reviewer that operationalization, particularly on quantifiable metrics that AI systems can optimize, is key. We want to note that the paper is not completely silent on this point: For instance, Section 5 draws on studies that employ quantifiable measures, including movement on attitudinal scales, cross-partisan donation decisions, endorsement of extreme comments, bridging scores that quantify cross-partisan agreement, and text analysis of perspective diversity in conversational transcripts. These measures are deployed in randomized experiments and field studies that we have reviewed and cited. Furthermore, the long literature on deliberation theory further offers established frameworks, including Steenbergen et al.'s (2003) Discourse Quality Index and Fishkin's (2021) measures of opinion change, knowledge gain, and preference coherence, which are not limited to AI interactions. In acknowledgement of Reviewer's point, we will reference these more explicitly in the next version of the paper. At the same time, we deliberatively call to action to the ML community to discuss further deliberative quality benchmarks as a priority, precisely because we view this as an open empirical question that a position paper should motivate.
>
> **AI capabilities and limitations.**
> We appreciate the concern and welcome the opportunity to clarify. Section 6 devotes four subsections to the limitations and risks of AI in deliberative contexts: sycophancy and the bias blind spot; training bias across ideological, cultural, demographic, and linguistic dimensions; over-reliance and cognitive offloading; and alignment failures including the systematic underrepresentation of minority perspectives in LLM-generated summaries. As stated in the paper, "AI systems designed to maximize user satisfaction will undermine deliberative goals" and LLMs' persuasive capacity creates "temptations to deploy AI for persuasion rather than augmentation." In that sense, our position is normative, not descriptive: we argue that AI systems *should* be designed to facilitate deliberation, not that current systems already do so reliably. We do not assume these tasks are solved. Rather, we argue they are worth solving, and we identify the specific challenges the ML community must address.
>
> **Evidence scope and at scale deliberation.**
> This is an important point, and we appreciate the opportunity to clarify how the evidence supports the position. Our argument proceeds in two steps. First, Section 5 establishes that each component of AI-assisted deliberation works in rigorous experimental settings: language assistance, AI-assisted reflection, comment recommendation, opinion visualization, and bridging-based consensus surfacing have each been tested in randomized or field experiments. Second, existing platforms already integrate subsets of these capabilities and have been deployed at substantial scale (e.g., Taiwan's vTaiwan engaged over 200,000 visitors; Decidim serves over 400 municipalities worldwide; deliberation.io caters thousands of residents in Washington D.C.).
>
> Moreover, the scalability of AI-assisted deliberation rests on specific, cost-effective mechanisms. As we describe in our response to Reviewer WGSB, we identify three specific areas: (1) Reflection, through even a one-round LLM conversation when eliciting users' reactions to contentious topics, implementable as a two-step process at small inference cost; (2) Synthesis, through automated summarization and opinion clustering over thousands of contributions, replacing teams of trained facilitators; and (3) Moderation, through AI-assisted enforcement of deliberative norms that would be prohibitively expensive with human moderators at tens of thousands of simultaneous participants. In each case, the marginal cost of serving an additional participant is negligible compared to the fixed costs of traditional deliberation (facilitators, venues, compensation). Again, we are not pushing for AI content moderation or summarization as an end, but for a pipeline where AI systems assist humans to reflect and identify areas of collaboration.
>
> We hope this clarifies how the evidence and scalability arguments support the paper's position. If there are specific additional points the Reviewer would like us to address, we would be happy to incorporate them in the revision.

---

> > ### Author Rebuttal · Reviewer_okNN · 2026-04-05
> >
> > The authors' response has addressed most of my previously raised concerns, and I would like to increase my overall rating.

---

### Official Review · Reviewer_WGSB · 2026-03-16

**Significance:** 3
**Argument Clarity:** 3
**Rating:** 4
**Confidence:** 4

**Questions:**

I hope to see the authors' perspective on the weaknesses mentioned above, for example: what Democratic Deliberation workflow this article adopts, how AI is specifically applied to Democratic Deliberation and in which part of the workflow it is applied, why the position emphasizes "at scale" and what distinguishes this from a general scenario, whether AI has particular advantages or limitations in an "at scale" scenario, and how to understand the connection between the Principles and the Behavioral Frictions in Figure 1.

**Alternative Views Section:**

Yes

**Compliance With Llm Reviewing Policy A Conservative:**

Affirmed.

**Discussion Potential:**

3

**Paper Summary:**

The authors hold the position: AI Should Facilitate Democratic Deliberation at Scale. The authors argue that AI systems, when designed to complement rather than substitute human judgment, offer an unprecedented opportunity to strengthen democracy by enabling deliberation at scale while addressing frictions that undermine meaningful civic engagement. The authors introduced four Behavioral Frictions in Online Deliberation (essential for designing AI systems and interventions that address root causes rather than symptoms) and identified four principles that should govern the integration of LLMs into deliberative platforms. For each Behavioral Friction challenge, the authors discussed the corresponding AI Capabilities. The authors also discussed Critical Challenges and Safeguards arising from AI for Democratic Deliberation. Finally, the authors discussed three Alternative Views and a Call to Action for researchers, platform developers, and policymakers.

**Position:**

Yes

**Position In Title:**

Yes

**Related Work:**

3

**Strengths And Weaknesses:**

The article's viewpoint is fairly novel and interesting, and the discussions of evidence, challenges, and call for action are also thorough.

Weaknesses lie in:

I feel Democratic Deliberation is a specialized and multi-step process that requires a definition or diagram to illustrate what modules are involved in this process and what work AI models perform within it. This is currently lacking in the article.

The position is "AI Should Facilitate Democratic Deliberation at Scale," but the article focuses mainly on discussing the advantages of AI for Democratic Deliberation without a dedicated discussion of the additional advantages and limitations of AI specifically in the context of "at scale." As we know, scalability is a concern in AI deployment due to high costs.

The article lacks a clear connection between the discussed Principles (Section 4) and the Behavioral Frictions from a societal perspective (Section 3), which makes the connection between the two in Figure 1 confused.

Typos:

Allen & Weyl, 2024 is cited multiple times repeatedly at line 186.

Line 309: Pol.is， the authors need to check the grammar here.

**Support:**

3

---

> ### Author Rebuttal · Authors · 2026-03-30
>
> We thank Reviewer WGSB for recognizing the novelty and thoroughness of our contribution and for the constructive suggestions.
>
> **Deliberation flow and the role of AI.**
> We thank and agree with the Reviewer that a visual representation of the deliberation flow would make the setting more clear. We want to note that the deliberation flow is embedded in the paper's structure: as shown in Sections 4 and 5, (1) input elicitation, (2) reflection, (3) structured exchange, and (4) synthesis are the areas where agreement may surface, and where AI may operate, guard-railed by the guiding principles listed in the paper. More specifically, Section 5 maps AI capabilities to these four stages: real-time language assistance; AI-assisted reflection; comment recommendation; opinion visualization; and bridging-based algorithms. As highlighted in the paper, the premise is that at no stage AI systems render judgments or generate direct conclusions. Rather, they operate by providing scaffolding to human reasoning and inputs, consistent with an augmentation principle. We will add a figure in the next version of the paper to make this flow and mapping explicit.
>
> **At scale deliberation.**
> We agree with the Reviewer that we should make the "at scale" dimension more evident in the paper. While we develop our main argument across four paragraphs in Section 2 (paragraphs 3-6), describing how the current versions of in-person or online deliberation are costly, we can include an additional paragraph describing concrete mechanisms through which AI can be scalable. We identify three specific cost-effective areas. First, Reflection, through engagement in (at least) one-round conversation with an LLM when eliciting users' initial reactions to contentious topics. This can be implemented as a two-step process, as current social media platforms already do (e.g., Instagram's "do you want to post this?" prompt: https://www.theguardian.com/technology/2019/jul/09/instagram-bullying-new-feature-do-you-want-to-post-this). Second, Synthesis, through automated summarization and opinion clustering over thousands of contributions, replacing what would otherwise require teams of trained facilitators to read, categorize, and distill. Platforms like Pol.is already perform this type of real-time clustering at scale. Third, Moderation, through AI-assisted enforcement of deliberative norms (civility checks, comment recommendation, equitable participation safeguards) that would be prohibitively expensive with human moderators at the scale of tens of thousands of simultaneous participants (as current content moderation teams on social media platforms, for example, https://www.bbc.com/news/articles/cgjyp48dp21o.). In each case, the marginal cost of serving an additional participant is negligible compared to the fixed costs of traditional deliberation (facilitators, venues, compensation), which is precisely what makes AI-assisted deliberation scalable. Again, we are not pushing for AI content moderation or summarization, but to a pipeline where the AI systems assist humans reflect and identify areas of collaboration.
>
> **Guiding principles as cross-cutting constraints.**
> The relationship between our four principles and the four friction categories is not intended as a one-to-one mapping. Section 4.1 presents this logic: "agency and equality establish who participates and on what terms, respect governs how participants engage, and augmentation constrains AI's role." Each principle governs *how* AI interventions should address frictions across all categories. Agency constrains all interventions to preserve human choice regardless of the friction being addressed; respect governs the character of AI-mediated interactions whether the underlying friction is cognitive, social, or structural; equality requires that no intervention exacerbate existing disparities in any domain; and augmentation prohibits substitution across all capabilities. We recognize that Figure 1 may unintendedly suggest a one-to-one correspondence that we do not intend, and we will revise the figure to depict principles as cross-cutting constraints.
>
> **Typos.**
> Thank you very much for the suggestion. We will remove the duplicate Allen & Weyl (2024) citation. Regarding line 309, we appreciate the note but want to clarify that the correct spelling of the platform is Pol.is, with an intermediate point, as per the platform's documentation (https://compdemocracy.org/Polis/).
>
> We hope this addresses the Reviewer's concerns. Please let us know if there are additional changes we can make to strengthen the paper.

---

> > ### Author Rebuttal · Reviewer_WGSB · 2026-04-05
> >
> > Thank you for the clarifications. My concerns have been adequately addressed, and I would like to maintain my current positive score.

---

### Decision · Program_Chairs · 2026-04-30

**Decision:**

Accept (spotlight)

**Comment:**

Given the overall positive attitude based on the significance of the problem and relatively well thought-out arguments, reviewers agreed that it is a clear acceptance.